# Reduced Graphene Oxide–Epoxy Grafted Poly(Styrene-Co-Acrylate) Composites for Corrosion Protection of Mild Steel

**Xinchuan Fan** [1,2] , **Yue Hu** [2] , **Yijun Zhang** [2] , **Jiachen Lu** [2] , **Xiaofeng Chen** [2] , **Jun Niu** [2] , **Ningyan Li** [2] **and Dianyu Chen** [1,2,*]

[1] School of Materials Science and Engineering, China University of Mining and Technology, Xuzhou 221116, Jiangsu, China; chuan@cumt.edu.cn

[2] Department of Chemistry & Materials Engineering, Jiangsu Key Laboratory of Advanced Functional Materials, Changshu Institute of Technology, Changshu 215500, Jiangsu, China; 20185209005@stu.suda.edu.cn (Y.H.); zyj52980576@163.com (Y.Z.); abc1476152338@163.com (J.L.); cslgcxf@163.com (X.C.); nj18020222290@163.com (J.N.); lny1106171643@163.com (N.L.)

* Correspondence: chendy@cslg.edu.cn; Tel.: +86-512-52251842; Fax: +86-512-52251842

**Abstract:** Reduced graphene oxide–epoxy grafted poly(styrene-co-acrylate) composites (GESA) were prepared by anchoring different amount of epoxy modified poly(styrene-co-acrylate) (EPSA) onto reduced graphene oxide (rGO) sheets through π–π electrostatic attraction. The GESA composites were characterized by Fourier transform infrared spectroscopy (FTIR), Raman spectroscopy, X-ray diffraction (XRD), scanning electron microscopy (SEM), transmission electron microscopy (TEM), and X-ray photoelectron spectroscopy (XPS). The anti-corrosion properties of rGO/EPSA composites were evaluated by electro-chemical impedance spectroscopy (EIS) in hydroxyl-polyacrylate coating, and the results revealed that the corrosion rate was decreased from $3.509 \times 10^{-1}$ to $1.394 \times 10^{-6}$ mm/a.

**Keywords:** reduced graphene oxide; poly(styrene-co-acrylate) composite; corrosion

## 1. Introduction

Graphene (G) and its family materials including graphene oxide (GO) and reduced graphene oxide (rGO) have been used widely for metal corrosion protection [1–4], which is beneficial due to its excellent physical barrier, chemical and electrochemical properties. However, because of strong van der Waals forces and high specific ratio, G, GO or rGO tend to easily aggregate each other, which greatly limits their functional development and industrial applications [5–7]. The chemical modification of G, GO or rGO is an efficient method to limit the aggregation, and at the same time, the anticorrosion performance of the nanocomposite coating is enhanced [8–19].

GO-PMMA(polymethyl methacrylate) composites were prepared by Chang in the presence of carboxyl compounds, which prevents the agglomeration of graphene in acrylate resin, and the anticorrosion property is twenty-seven times higher than that of pure PMMA film [20]. Polyaniline/4-aminobenzoic acid was also employed by Chang to modify graphene, and the PANI(polyaniline)–G composite with 0.5 wt.% content can reduce the diffusion rate of water and oxygen to 22% and 24% [21]. Yu modified graphene by phenylenediamine and 4-vinylbenoic acid, and the pvGO(Polyvinyl alcohol-graphene oxide) composite can be dispersed in water, and the corrosion protection efficiency (2 wt.% of pvGO) in polystyrene coating is increased from 37.90% to 99.53% [22]. A cationic rGO (rGO–ID$^+$) composite was prepared by modifying rGO with isophorone diisocyanate $N, N$-dimethylethanolamine, and the anticorrosion property of which has been significantly improved (corrosion potential ($E_{corr}$) reaches $-0.302 \pm 0.004$ V/SCE, and corrosion current density ($i_{corr}$) is at

only $3.141 \times 10^{-4}$ μA/cm$^2$) [23]. However, up to now, the G, GO or rGO composites modified with polyacrylate derivatives for anticorrosion properties have rarely been reported, which have potentially wide industrial application.

Herein, we synthesized graphene oxide-epoxy grafted poly(styrene-co-acrylate) composites (GESA) by anchoring epoxy modified poly(styrene-co-acrylate) (EPSA) resin onto rGO sheets through π–π electrostatic attraction. Modification conferred rGO sheets with uniform dispersion in the polymer matrix because of the good compatibility of acrylate and epoxy units in hydroxyl polyacrylate resin. The results proved that the modified rGO sheets could be used as a barrier fillers in hydroxyl polyacrylate resin coating and the corrosion protection of GESA was more effective than that of unmodified rGO sheet. The GESA composites were characterized by FTIR, Raman spectra, XRD, SEM, TEM, and X-ray photoelectron spectroscopy (XPS). The corrosion protection of GESA composites coating were studied using electro-chemical impedance spectroscopy (EIS).

## 2. Experimental

### 2.1. Materials

Benzoyl peroxide (BPO, Macklin), methyl acrylate (MA, Macklin), methyl methacrylate (MMA, Macklin), methacrylic acid (MAA, Macklin), butyl acrylate (BA, Macklin), hydroxyethyl methacrylate (HEMA, Macklin), styrene (St, Macklin), *N*-methyl pyrrolidinone (NMP, Macklin), ethyl hexanoate (SG, Macklin), 1-dodecanethiol (NDM, Macklin), acetone (Macklin), xylene (Macklin) and butyl acetate (Macklin) were purchased from Macklin (Newport Beach, CA, USA). E-20 (molecular weight = 2000 g/mol) was purchased from Sinopec (Beijing, China). All of these reagents were used directly without further purification. Reduced graphene oxide was purchased from the Sixth Element Materials Technology Co., Ltd. (Changzhou, China). Hydroxyl polyacrylate (BS-963) was provided from Sanmu Group Co., Ltd. (Yixing, China).

The mild steel panels was a kind of steel with the following composition: 0.050 wt.% S, 0.45 wt.% Mn, 0.30 wt.% Si, 0.045 wt.% P, 0.18 wt.% C, balance Fe. The mild sheet with a dimension of 3 cm × 5 cm was used as the substrate for corrosion evaluation. These panels were degreased with ethanol/butyl acetate solution prior to film application.

### 2.2. Synthesis of EPSA

Before reaction, a raw material solution was mixed as following composition: 1 g of methyl acrylate, 1 g of methyl methacrylate, 1 g of hydroxyethyl methacrylate, 2 g of methacrylic acid, 10 g of butyl acrylate, 72 g of styrene, 2 g of BPO, 0.3 g of 1-dodecanethiol, and 50 g of NMP. The free radical polymerization occurred in a 250 mL round-bottomed flask with water separator and mechanical stirrer, under nitrogen atmosphere in an oil bath at 140 °C. The mixed solution was dropped into the flask slowly, and continuously reacted for 4 h. Then, 88 g of E-20 and 12 g of caproic acid were added and refluxed for another 3 h, and the byproduct of water was exported via the water separator. The sticky polymer product was obtained with faint yellow color.

### 2.3. Preparation of rGO/EPSA Composites

rGO (2 g) and NMP (20 g) were added into a flask and dispersed in an ultrasound bath for 30 min, then EPSA resin (0.5 g, 1 g, 1.5 g, 2 g, and 3 g for GESA1, GESA2, GESA3, GESA4, and GESA5, respectively) was added and stirred for 90 min. The resulting GESA suspension was rinsed with deionized water and acetone, filtered, and dried under vacuum at 60 °C to a dry powder.

### 2.4. Preparation of the Coating

In this research, the samples consisted of (1) neat resin (hydroxyl polyacrylate, HPA) containing no rGO and EPSA, (2) resin with 0.4 wt.% rGO (rGO/HPA), and (3) resin with GESA containing

0.4 wt.% rGO (GESA/HPA). These samples were used to prepare coatings. Three kinds of coatings were prepared by the following processes:

HPA: The hydroxyl polyacrylate resin was dissolved in a stoichiometric solvent (a mixture of xylene and butyl acetate in a weight ratio of 1:1). The mixture was held in a vacuum chamber at 20 °C for 20 min to remove air bubbles, which were then applied to the pretreated mild steel sheet.

rGO/HPA: The calculated amounts of rGO powder was dispersed with solvent and ultrasonic processed for 30 min to obtain a homogeneous dispersion. The hydroxyl polyacrylate resin was then added into the dispersion. The mixture was added into an agate jar of 50 mL inner volume and then ground for 90 min (agate grinding ball to the mixture with weight ratio of 1:1) at 600 r/min. The ball milled mixture was held in a vacuum chamber at 20 °C for 20 min to remove air bubbles, which were then applied to the pretreated mild steel sheet.

GESA/HPA: GESA composites, hydroxyl polyacrylate resin, and solvent were blended in a flask under stirring. The mixture was added into an agate jar of 50 mL inner volume and then ground for 90 min (agate grinding ball to the mixture with weight ratio of 1:1) at 600 r/min. The ball milled mixture was held in a vacuum chamber at 20 °C for 20 min to remove air bubbles, which were then applied to the pretreated mild steel sheet.

The coating was dropped onto the substrates and prepared for uniform coating by using a coating applicator. The surfaces of all selected specimens should be flat without defects, such as pores or blisters. The thickness of each dry coating sample was 50 ± 5 μm.

## 2.5. Characterization

FTIR spectra were recorded using a Nicolet IS50 spectrophotometer (Thermo Fisher scientific, Beijing, China) in KBr pellets. The Raman spectra of the composites were obtained using a Takram-P50C0R10 Raman microscope spectrometer (Bruker, Karlsruhe, Germany) at the excitation of a 532 nm Nd:YAG(Yttrium Aluminum Garnet) laser. XRD measurements were performed by an APEX II DUO diffractometer (Bruker, Karlsruhe, Germany) with CoKα radiation ($\lambda$ = 1.7890 Å). The morphology of the rGO sheet and GESA composites were characterized by SEM (SUPRA55, Carl Zeiss, Munich, Germany). For improving contrast, rGO sheet and GESA composites were pretreated with gold sputtering. TEM images of the samples were obtained on a JEM-2100 transmission electron microscope (TEM, Tokyo, Japan) at 200 kV. XPS was carried out with a Perkin–Elmer PHI-5702 multi-technique spectrometer (Physical Electronics, Chanhassen, MN, USA), using AlKα excitation radiation.

The corrosion protection of the coatings was tested using EIS measurements in a conventional three-electrode cell on an RST5500 Electrochemical Workstation (Dufu Instrument, Zhengzhou, China). The coated mild steel was used as the working electrode. A large platinum plate was used as the counter electrode, and the Ag/AgCl (saturated KCl) electrode was used as reference electrode. Measurements were performed on 2 cm$^2$ coatings immersed in 3.5% NaCl solution. All tests were conducted at room temperature (approximately 20 °C) and exposed to air.

EIS measurements were performed within the frequency range of 100 kHz to 1 Hz by using a 30 mV amplitude sinusoidal electrode potential at the open-circuit potential (OCP). Data were acquired in four cycles at each frequency to obtain good precision at all frequencies. The electrochemical parameters of each coated system were measured from three independent specimens, and the results show the mean and standard deviation.

## 3. Results and Discussion

### 3.1. Characterization of GESA

The chemical changes that occurred during treatment of rGO with EPSA were observed by FTIR spectroscopy. The FTIR spectra of rGO, GESA1, GESA2, GESA3, GESA4, GESA5, and EPSA are shown in Figure 1. Absorption peak intensity of organic groups increased with increasing EPSA. Compared with GESA5 and EPSA, the new peaks were not found, and the disappearing of 1666 and 2072 cm$^{-1}$ (arene ring

vibration) in GESA might be caused by the electron delocalization from graphene aromatic ring to arenes of EPSA; additionally, the characterized peaks of rGO was not changed from GESA1 to GESA5. It might be deduced that no covalent reaction but only π–π electrostatic attraction ocurred between rGO and EPSA.

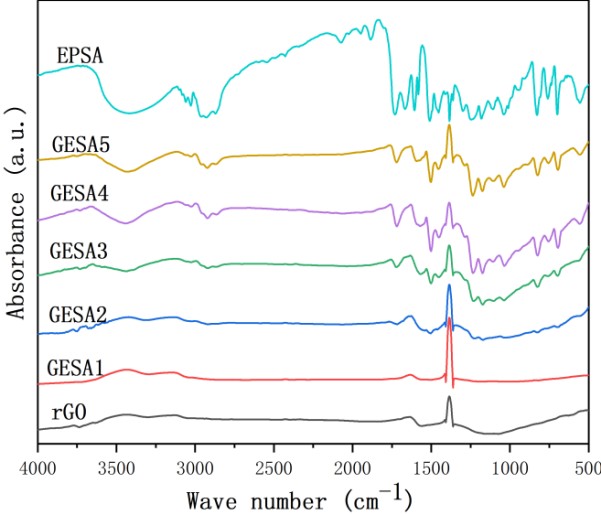

**Figure 1.** FTIR spectra of reduced graphene oxide (rGO), graphene oxide-epoxy grafted poly(styrene-co-acrylate) composites (GESA)1, GESA2, GESA3, GESA4, GESA5, and epoxy modified poly(styrene-co-acrylate) (EPSA).

The XRD diffraction patterns of the rGO, GESA1, GESA2, GESA3, GESA4, and GESA5 are shown in Figure 2. rGO had a characteristic diffraction peak associated with the (001) plane at 2θ = 25.87°, and exhibited a highly ordered structure according to Bragg's law ($n\lambda = 2d\sin\theta$; $\lambda = 0.1789$ nm), [24,25] and the layer spacing (*d*) was about 3.44 Å. For GESA1, GESA2, GESA3, GESA4, and GESA5 samples, the XRD pattern showed that the (001) characteristic peaks moved to lower 2θ values at 25.78°, 25.66°, 25.34°, 25.22°, and 24.86° respectively, which might have been caused by the intercalation of EPSA between rGO nanosheets. The appearance of new peaks at 19.23°, 18.71°, and 18.67° (for sample GESA3, GESA4, and GESA5) might have been attributed to the arene arrangement of excess ESPA.

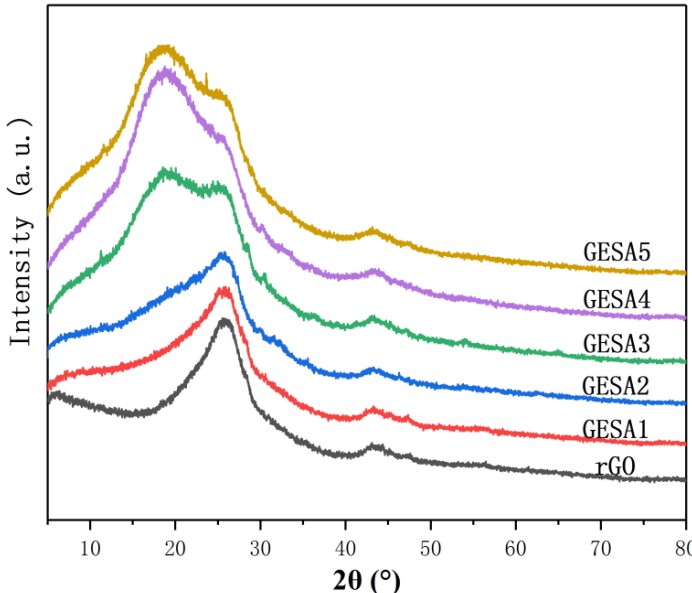

**Figure 2.** XRD diffractograms of rGO, GESA1, GESA2, GESA3, GESA4, and GESA5 composites.

Raman spectroscopy is considered to be a powerful analytical technique for identifying defects and disorder levels of graphene-based structures [26]. The Raman spectra of rGO, GESA1, GESA2, GESA3, GESA4, and GESA5 are shown in Figure 3, and two characteristic D and G peaks were observed for all samples. The D band arose from the defects and disorder of carbon in graphene platelets and moreover, the G band originated from the first-order scattering of the $E_{2g}$ vibration mode and the in-plane vibration of ordered $sp^2$-bonding carbon atoms [27]. The peak intensity ratio of the D band to the G band ($I_D/I_G$) was investigated to examine the degree of defects and disorder. With EPSA increasing, the $I_D/I_G$ values of GESA1, GESA2, and GESA3 were reduced from 1.44 to 1.12, 1.05, 0.67 respectively. The degree of defect and disorder of GESA was slightly lower than that of rGO for the samples of GESA1, GESA2, and GESA3, which was inconsistent with literatures, might have been caused by the ordered arrangement of arenes of EPSA on the surface of rGO (just as explained in IR and XRD analysis). With increasing EPSA, the excess ESPA widened the distance of rGO nanosheet, the degree of defect and disorder of GESA gradually increased, and the $I_D/I_G$ value of GESA4 and GESA5 increased conversely to 1.41 and 1.54.

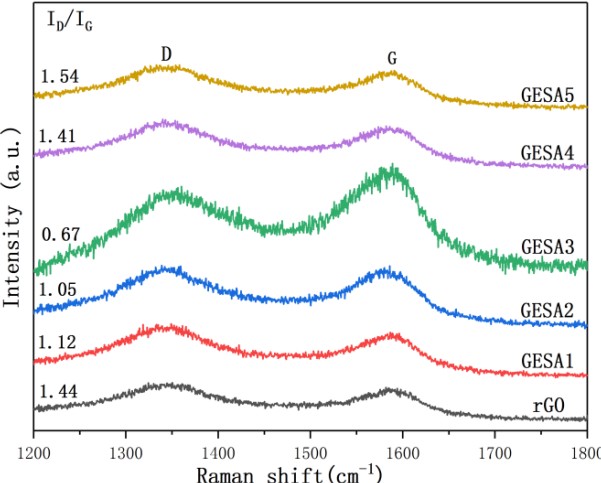

**Figure 3.** Raman spectra of rGO, GSA1, GESA2, GESA3, GESA4, and GESA5 composites.

The morphologies and microstructures of the five GESA composites and rGO sheets were studied by SEM (Figure 4). rGO exhibited nanoscale sheets with wrinkles and a wavy feature (Figure 4a), the interconnection between nanosheets portended an agglomeration trending with application. With increasing EPSA, the wrinkles of the GESA composites disappeared (Figure 4b–f), and the edge contour became clearer and clearer.

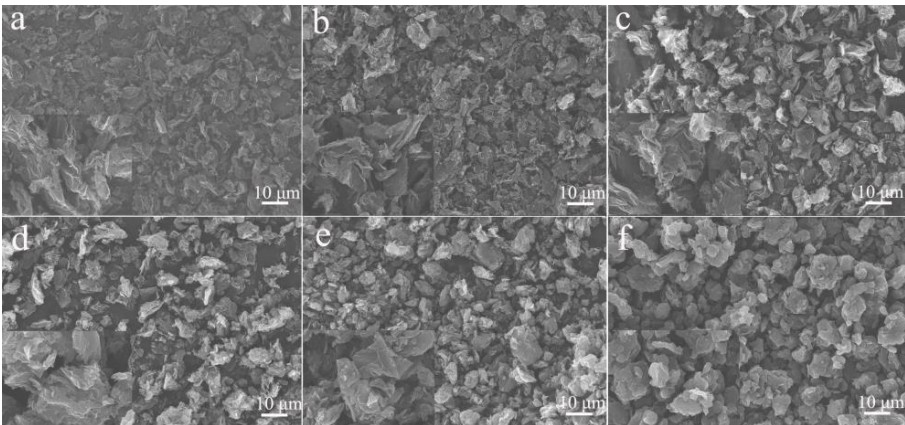

**Figure 4.** SEM images of rGO (**a**), GESA1 (**b**), GESA2 (**c**), GESA3 (**d**), GESA4 (**e**), and GESA5 (**f**) composites.

Further, the TEM image (Figure 5) showed a high degree of silk-like structure in both rGO sheet and GESA composites. It could be seen from Figure 6a that rGO was agglomerated and intertwined with each other. While GESA was loosened after adding EPSA, the sheets were disaggregated (Figure 5b–f), and the black area could be observed among silks, which was attributed to self-arranged EPSA between rGO.

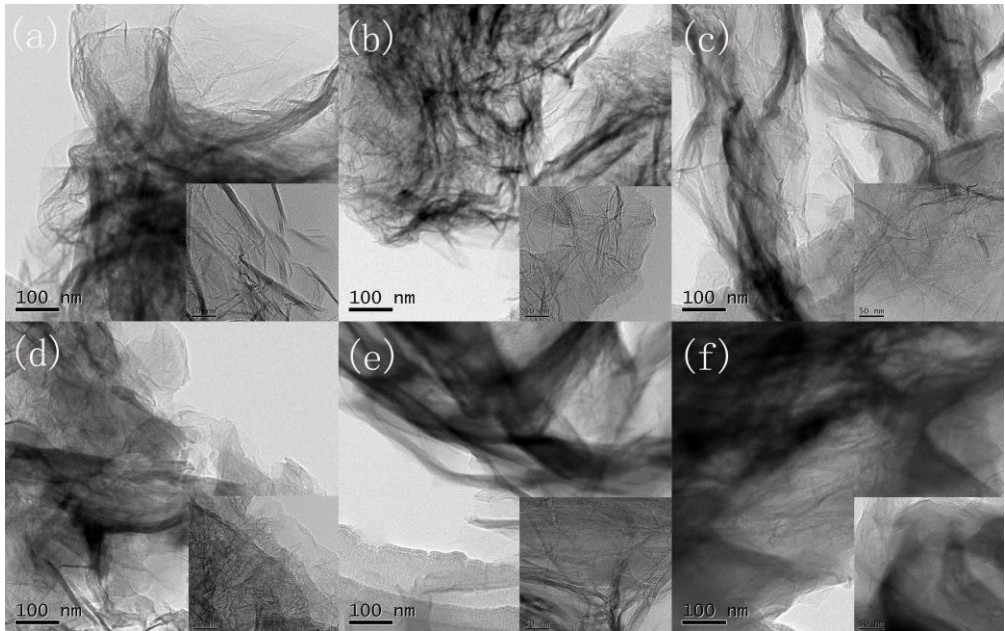

**Figure 5.** TEM images of rGO (**a**), GESA1 (**b**), GESA2 (**c**), GESA3 (**d**), GESA4 (**e**), and GESA5 (**f**) composites.

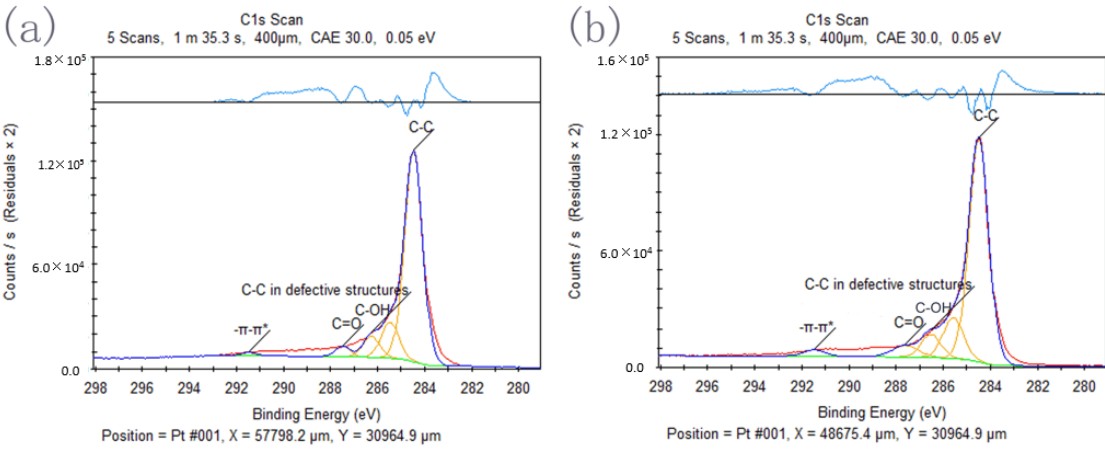

**Figure 6.** C1*s* X-ray photoelectron spectroscopy (XPS) spectra of rGO (**a**) and GESA3 (**b**).

XPS measurements were performed on rGO and GESA3 samples, and the high-resolution C1*s* spectra are shown in Figure 6 and Table 1. In the case of rGO, the five peaks at 284.48, 285.48, 286.28, 287.50, and 291.50 eV represented C–C, C–C in defective structures, C–OH, C=O and –π–π * respectively [28,29]. Compared with rGO, almost no changes in peak position was found for the C1*s* XPS spectrum of GESA3. Noteworthily, the peak area ratio of –π–π * was greatly increased from 0.93% to 2.08%, which might have been caused by electron attraction of arenes in EPSA. Therefore, we confirmed that not it was not the chemical bond but the π–π electrostatic attraction that occurred between rGO and EPSA during the modification process. A deduction was established that the electron attraction was the anchoring force between arene rings (in EPSA) and rGO, which ensured the stability of GESA [30,31].

**Table 1.** Peak and area of rGO and GESA3.

| Name | | C–C | C–C in Defective Structures | C–OH | C=O | −π–π* |
|---|---|---|---|---|---|---|
| Peak (eV) | rGO | 284.48 | 285.48 | 286.28 | 287.50 | 291.50 |
| | GESA3 | 284.50 | 285.58 | 286.53 | 287.63 | 291.50 |
| Atomic | rGO | 76.47 | 12.61 | 7.09 | 2.90 | 0.93 |
| | GESA3 | 72.12 | 13.63 | 7.22 | 4.95 | 2.08 |

According to these characterizations, the possible schematic of GESA composites is shown in Scheme 1.

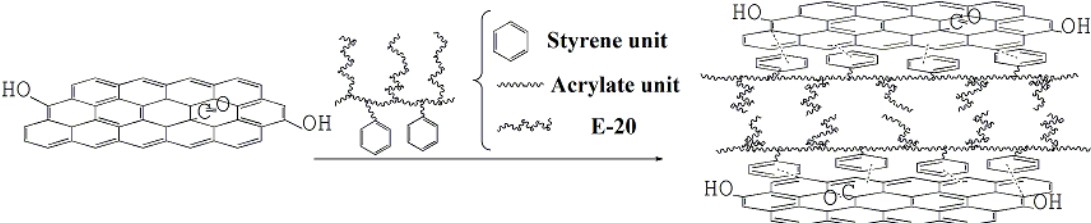

**Scheme 1.** The synthesis schematic of the GESA composites.

## 3.2. Corrosion Studies

The protection efficiency and corrosion rate of GESA coated mild steel were evaluated from Tafel analysis. The corrosion mechanism of mild steel can be attributed to the anodizing process occurring at the working electrode and the reduction process at the cathode, which utilized electrons released during the oxidation of Fe. The different types of anode and cathode reactions that occur on each electrode are given below [32,33].

At the cathode:

$$O_2 + 2H_2O + 4e^- \rightarrow 2OH + 2OH^- \tag{1}$$

$$OH + e^- \rightarrow OH^- \tag{2}$$

At the anode:

$$Fe + 2OH^- \rightarrow Fe(OH)_2 \tag{3}$$

$$2Fe(OH)_2 \rightarrow Fe_2O_3 \cdot XH_2O \tag{4}$$

At the cathode, a decrease in oxygen occurred with the consumption of electrons released at the anode, which facilitated the formation of hydroxide ions in the electrolyte medium. Then, the obtained hydroxide ion was easily combined with Fe produced on the anode to obtain iron hydroxide, which was further converted into iron oxide [33]. All of these reactions occurred simultaneously. If any external process limited anodization or prevents $OH^-$ ions, it could inhibit the corrosion of Fe.

The potentiodynamic polarization test was performed after a stable OCP measurement was reached. The anode and cathode polarization curves recorded for the HPA coating and GESA coated mild steel in 3.5% NaCl solution are shown in Figure 6 for bare mild steel. The corrosion current density ($i_{corr}$) obtained from the Tafel curve determined the protection efficiency and corrosion rate of the HPA coating and the GESA coated mild steel. The measured $i_{corr}$ and $E_{corr}$ values are summarized in Table 2.

| Specimen | Corrosion Potential, $E_{corr}$ (V) | Corrosion Current Density, $i_{corr}$ (A) | Corrosion Rate, $v_{corr}$ (mm/a) |
|---|---|---|---|
| Bare mild steel | −0.9178 | $2.980 \times 10^{-5}$ | $3.509 \times 10^{-1}$ |
| HPA coated mild steel | −0.7115 | $2.911 \times 10^{-8}$ | $3.427 \times 10^{-3}$ |
| rGO/HPA coated mild steel | −0.7382 | $9.216 \times 10^{-9}$ | $1.085 \times 10^{-3}$ |
| GESA1/HPA coated mild steel | −0.5187 | $4.629 \times 10^{-10}$ | $5.450 \times 10^{-6}$ |
| GESA2/HPA coated mild steel | −0.4792 | $2.147 \times 10^{-10}$ | $2.528 \times 10^{-6}$ |
| GESA3/HPA coated mild steel | −0.4006 | $1.184 \times 10^{-10}$ | $1.394 \times 10^{-6}$ |
| GESA4/HPA coated mild steel | −0.4182 | $1.782 \times 10^{-10}$ | $2.098 \times 10^{-6}$ |
| GESA5/HPA coated mild steel | −0.6179 | $2.568 \times 10^{-9}$ | $3.023 \times 10^{-5}$ |

The $i_{corr}$ value of mild steel was $2.980 \times 10^{-5}$ A, the $i_{corr}$ value was decreased to $2.911 \times 10^{-8}$, $9.216 \times 10^{-9}$, $4.629 \times 10^{-10}$, $2.147 \times 10^{-10}$, $1.184 \times 10^{-10}$, $1.782 \times 10^{-10}$, and $2.568 \times 10^{-9}$ A, after coating with HPA, rGO/HPA, and GESA/HPA. The $E_{corr}$ values (with the same orders as $i_{corr}$) measured from Tafel plots (Figure 7a) were −0.9178, −0.7115, −0.7382, 0.5187, −0.4792, −0.4006, −0.4182, and −0.6179 V. Compared with the bare mild steel, the $E_{corr}$ value of all samples decreased to some extent, and the maximum decline of 517.2 mV was seen for GESA3. The significant shift in $E_{corr}$ value along with the reduction in $i_{corr}$ values showed that the GESA/HPA films possessed better performance in 3.5% NaCl solution thus preventing the interaction or the direct contact of ions present in corrosive medium with the metal substrate and acted as an excellent protective barrier [34].

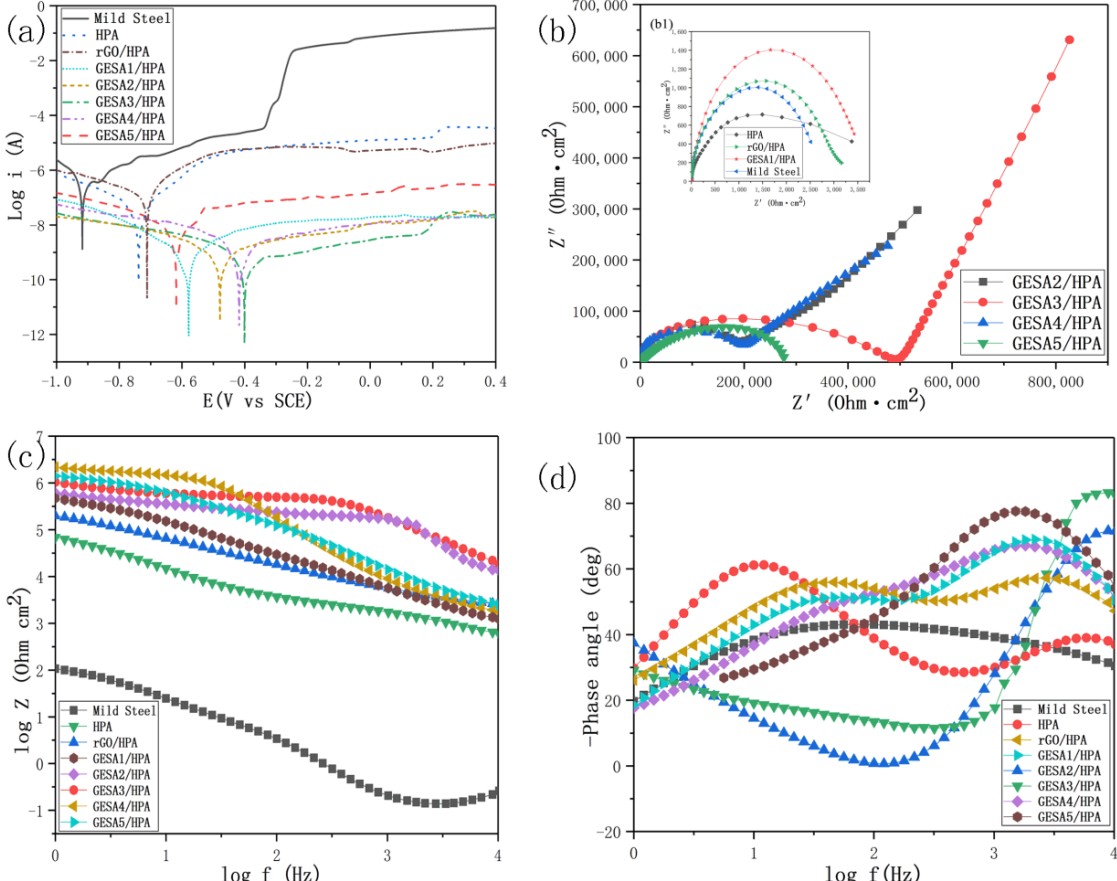

**Figure 7.** (**a**) Tafel plots, (**b**) Nyquist plots, (**c**) Bode modulus plots, and (**d**) Bode phase plots of uncoated and coated mild steels in 3.5 wt.% NaCl solution.

The corrosion rate $v_{\text{corr}}$ in mm/a is calculated by assuming uniform penetration rate of chloride ions onto the bare mild steel, HPA, rGO/HPA, and GESA/HPA coated mild steel using the following expression [35]:

$$v_{\text{corr}} = \frac{i_{\text{corr}} K E_w}{\rho A} \tag{5}$$

where $E_w$ is the equivalent weight of mild steel, $K$ is constant equal to $1.288 \times 10^5$, $\varrho$ is the density of the mild steel and $A$ is the area of specimen exposed to the corrosive medium. The obtained corrosion rate is also given in Table 1.

From Figure 7a and Table 1, it is clearly inferred that GESA coated mild steel exhibited better corrosion resistance than the bare mild steel, HPA coated mild steel and rGO/HPA coated mild steel specimens. The corrosion rate of the bare mild steel was $3.509 \times 10^{-1}$ mm/a, which was reduced to $3.427 \times 10^{-3}$ and $1.085 \times 10^{-3}$ mm/a on coating HPA and rGO. Covered with a further reduction in the corrosion rate of the GESA coating, where coated GESA3 was reduced to $1.394 \times 10^{-6}$ mm/a. The Nyquist plot, Bode modulus plot, and Bode phase plot of uncoated and coated mild steels are presented in Figure 7b–d.

## 4. Conclusions

In summary, five GESA composites with different mass ratio (from 1:0.25 to 1:1.5) were synthesized. The EPSA resin was modified onto rGO sheets through π–π electrostatic attraction. Modification conferred rGO sheets with uniform dispersion in the polymer matrix because of the good compatibility of acrylate and epoxy units in organic coatings. Experimental results proved that the reduced graphene oxide-epoxy grafted poly(styrene-co-acrylate) composites could enhance the barrier property of hydroxyl polyacrylate coating. The GESA3 composites conferred optimal corrosion protection, the corrosion rate tested by EIS on mild steel plates was decreased from $3.509 \times 10^{-1}$ to $1.394 \times 10^{-6}$ mm/a.

**Author Contributions:** Conceptualization, D.C.; Methodology, Y.Z. and J.N.; Software, N.L.; Investigation, Y.H.; Writing—original draft preparation, X.F.; writing—review and editing, X.F. and D.C.; Visualization, J.L.; Supervision, X.C.

**Funding:** This research received no external funding.

**Conflicts of Interest:** The authors declare no conflict of interest.

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
