# Peer review of "Reduced Graphene Oxide–Epoxy Grafted Poly(Styrene-Co-Acrylate) Composites for Corrosion Protection of Mild Steel"

_coatings, doi:10.3390/coatings9100666_

Round 1

Reviewer 1 Report

The authors present a study on corrosion protection of mild still using reduced graphene oxide-epoxy grafted poly(styrene-co-acrylate) composites. The improvement in the corrosion rate is impressive enough to draw potential readers’ attention. However, some unclear discussion does not persuade the reason for the improvement. After minor revision of the manuscript and English editing, it could be considered for publication in Coatings.

- The authors argue that many features in spectra (Figures 1, 2, and 3) were caused by the arrangement of arenes of EPSA (line 129-130, line 142-143, and line 155-157). They need references to be supported.

- In Figure 4, I agree that the average size of particles is getting bigger and losing their sharp edges. However, I do not understand what the microcake structures mean in line 168.

- What is the silk-like structure in Figure 5? In addition, I cannot see any noticeable difference between TEM images. To show agglomerates, low magnification images should be provided. Otherwise, there is only the wrinkled structure of GOs without disctinction among them.

- Table 1 need correction. The C-OH, C=O, -pi-pi* are not atoms, so the expression ‘atomic %’ is wrong.

- The list of authors is not given properly. Clarify who is the corresponding author.

- There are some minor grammatic errors or typing errors. For example, in line 25, ‘G, GO, or rGO’ is a better expression than ‘G or GO or rGO’. An awkard expression like line 40 should be corrected. The authors need to check the entire manuscript.

- Some abbreviations are defined more than once. To name a few, FTIR, XRD, TEM, XPS in section 2.5 are examples. There were already defined at the end of introduction. At the same time, there are abbreviations which were not defined in the manuscript. For example, pvGO (line 34) and Ecorr and icorr (line 38), EPSA (line 42) should be defined.

- In section 2.1, the materials used in this study were capitalized. I do not think it is necessary. On the other hand, the first letter should be capitalized in the titles of 2.2, 2.3, and 2.4.

- In section 2.2, in line 68, replace grams by g for consistency.

Author Response

- The authors argue that many features in spectra (Figures 1, 2, and 3) were caused by the arrangement of arenes of EPSA (line 129-130, line 142-143, and line 155-157). They need references to be supported.

Answer: New reference of [28, 29] have been added in the manuscript.

- In Figure 4, I agree that the average size of particles is getting bigger and losing their sharp edges. However, I do not understand what the microcake structures mean in line 168.

Answer: The morphology in SEM looks like microcakes.

- Table 1 need correction. The C-OH, C=O, -pi-pi* are not atoms, so the expression ‘atomic %’ is wrong.

Answer: “%”have been deleted.

- The list of authors is not given properly. Clarify who is the corresponding author.

Answer: The corresponding author of “Dianyu Chen” have been marked.

- There are some minor grammatic errors or typing errors. For example, in line 25, ‘G, GO, or rGO’ is a better expression than ‘G or GO or rGO’. An awkard expression like line 40 should be corrected. The authors need to check the entire manuscript.

Answer: We have modified the proposed grammatic errors, and have checked the entire manuscript.

- Some abbreviations are defined more than once. To name a few, FTIR, XRD, TEM, XPS in section 2.5 are examples. There were already defined at the end of introduction. At the same time, there are abbreviations which were not defined in the manuscript. For example, pvGO (line 34) and Ecorr and icorr (line 38), EPSA (line 42) should be defined.

Answer: The abbreviation errors have been revised.

- In section 2.1, the materials used in this study were capitalized. I do not think it is necessary. On the other hand, the first letter should be capitalized in the titles of 2.2, 2.3, and 2.4.

Answer: The first letters in the titles have been capitalized.

- In section 2.2, in line 68, replace grams by g for consistency.

Answer: “grams” have been revised as “g”.

Reviewer 2 Report

The manuscript entitled "Reduced graphene oxide-epoxy grafted poly(styrene-co-acrylate) composites for corrosion protection of mild steel'' reports on the preparation of RGO/epoxy grafted poly(styrene-co-acrylate) composites by linking the epoxy modified poly(styrene-co-acrylate) onto RGO through π-π interactions and its beneficial role against corrosion as barrier in hydroxyl polyacrylate resin coatings.

The English of the manuscript is extremely poor and needs editing. On the other hand, novelty is not clearly presented compared to other similar studies. The prepared composites are adequately characterized. I recommend its publication to Coatings after major revision.

For the publication, the authors have to address my comments, as described below.

Please check author names. Who is and *? Graphene and its derivatives are not dissolved but dispersed! Please correct it in the manuscript. TGA measurements are required.

Author Response

Please check author names. Who is and *? Graphene and its derivatives are not dissolved but dispersed! Please correct it in the manuscript. TGA measurements are required. 

Answer: The corresponding author of “Dianyu Chen” have been marked. “dissolved” have been revised as “dispersed”.

Reviewer 3 Report

Experiments can be extended by observing the cross-section of the tested samples - to examine the quality of steel contact with the coating layer.

Extended Conclusions would be an advantage (e.g. what causes GESA3 the best; what about other materials - a general trend).

Fig. 4 could be improved (contrast).
Fig. 6 could be enlarged and improved (contrast)

Author Response

Experiments can be extended by observing the cross-section of the tested samples - to examine the quality of steel contact with the coating layer.

Extended Conclusions would be an advantage (e.g. what causes GESA3 the best; what about other materials - a general trend).

Answer: Thanks! All of these ideas will be considered in our following works.

Fig. 4 could be improved (contrast).

Answer: Fig. 4 have been improved as our understanding.
Fig. 6 could be enlarged and improved (contrast)

Answer: Fig. 6 cannot be enlarged in our software.